# OpenReview forum: "From CLIP to DINO: Visual Encoders Shout in Multi-modal Large Language Models"
_ICLR.cc/2024/Conference — Submitted to ICLR 2024_

### Official Review · Reviewer_JjPW · 2023-10-28

**Soundness:** 2 fair
**Presentation:** 1 poor
**Contribution:** 2 fair
**Rating:** 3
**Confidence:** 5

**Summary:**

This paper explores the effectiveness of various vision encoders in Multi-modal Large Language Models (MLLMs). While current MLLM implementations often employ CLIP or its variants for visual understanding and predominantly extract features from the model's deep layers, there is a lack of in-depth analysis of these visual encoders. Through an extensive examination, the authors discover that the shallow layer features of CLIP are especially beneficial for detailed tasks like grounding and region comprehension. Intriguingly, the DINO model, which is not initially trained with text-image alignment, shows significant potential as a visual component in MLLMs. When supplemented with an MLP layer for alignment, DINO outperforms CLIP in tasks demanding finer perception. Based on these insights, the paper introduces a novel feature integration method called COMM, which combines the strengths of both CLIP and DINO via Multi-level features Merging, aiming to enhance the visual proficiency of MLLMs. Evaluations of COMM on multiple benchmarks, such as image captioning, visual question answering, visual grounding, and object hallucination, indicate its superior performance over current methods, highlighting its improved visual competency in MLLMs.

**Strengths:**

1.	The paper introduces a novel strategy COMM that combines multi-stage visual features from both CLIP and DINO, attaining noticeable improvements on several VL reasoning tasks.
2.	The paper presents an in-depth investigation into the correlations among token-level visual features of vision transformers, revealing the issues of overlooking detailed local features in CLIP. This exploration may give very useful insights into the studies of representation learning.
3.	It also provides a comprehensive comparison of various candidate vision encoders that roughly cover the prominent pretraining strategies such as CLIP, DINO, MAE, and supervised pretraining (DEiT). The findings of MAE and DEiT’s failure to adapt to multi-model tasks are interesting, if the experiments are correctly implemented.

**Weaknesses:**

1.	My major concern lies in the contributions of this work. The new approach COMM seems more like an ensembling strategy to directly combine CLIP and DINO’s multi-stage visual features. The motivation demonstrated in the paper to introduce DINO features is not really convincing: despite the fact that DINO yields better localized features than CLIP, there exist a lot of simple solutions to overcome this problem such as re-organizing CLIP’s self-attention [1] and learning additional adaptation layers [2]. How would the difference be between your method and such counterparts?
2.	Also, merging multi-layer visual features in vision transformers does not necessarily make sense, because in such models there already exist residual connections between neighboring blocks which means features have already been merged to some extent. Will COMM still be robust if merging the features with an interval (e.g., layer 1, 3, 5, 7…)? This ablation study should be included.
3.	The experimental results do not seem very significant. If my understanding of your method is correct, you by default employ both CLIP and DINO for training and inference. However, your method demonstrates 0.3 lower avg. result than that with single DINO model in the “REC” task and obtains the same results in the absence of DINO in the “POPE” task.
4.	A minor concern comes from the dependence of a well-pretrained DINO. I understand that MAE-like and supervised models cannot achieve the performance you expect, but there are also many contrastive learning-based pretraining models such as MoCo v3, BYOL, as well as iBOT which derives from DINO and shares similar features with it. Is DINO the optimal choice of your method? If so, you should explain why it significantly outperforms other models.
5.	Minor comments: overall, the writing of this paper should be improved. Some important details are hard to understand. For example, the authors say “Previous MLLMs usually utilize…” in the first sentence of Section 3. However, there is no clear reference to indicate which MLLMs did. Such expressions might make readers really confused.

[1] Zhou, Chong, Chen Change Loy, and Bo Dai. "Extract free dense labels from clip." ECCV, 2022.

[2] Gao, Peng, et al. "Clip-adapter: Better vision-language models with feature adapters." IJCV, 2023.

**Questions:**

See "Weaknesses".

---

### Official Review · Reviewer_JffD · 2023-10-30

**Soundness:** 2 fair
**Presentation:** 3 good
**Contribution:** 2 fair
**Rating:** 3
**Confidence:** 5

**Summary:**

The paper investigates different visual encoders in the framework of multimodal LLM. Based on the observation, where shallow layers may provide more fine-grained semantics, the paper introduces a multi-level feature fusion strategy to improve the visual representation. Furthermore, the authors integrate the features of DINOv2 and CLIP-ViT for better results. The proposed framework with fused visual features demonstrates SOTA results on some public benchmarks.

**Strengths:**

+ The paper is easy to follow.
+ It is helpful to awaken everyone's attention to visual encoders in the era of LLM.
+ The effectiveness of the introduced method has been proven on some benchmarks.

**Weaknesses:**

-	The quantitative comparison in Table2-5 are unfair. The baseline MLLMs are trained with 224x224 images, while COMM was trained was 336x336 images. It is thus questionable if the higher performance comes from using high resolution images. The performance of the model trained with 224x224 images should also be provided for fair comparison.


-	The evaluations in this paper is insufficient. A more thorough comparison in various tasks such as those in [1,2] and recent MLLM benchmarks [3,4] should be provided to comprehensively compare existing methods. Furthermore, the evaluation in LLaVa[5] should be provided to see how much the model is as good as GPT-4.



-	The novelty of this work is limited. This paper claims itself to be the first work to study visual encoders for MLLM. However, existing work [6] has already provided similar investigations. On the other hand, the design of adapting multi-level CLIP features has also been proposed [7]. But the paper lacks discussion and comparison with these works.


-	The authors provide experiments with a single type of LLM, i.e., Vicuna. It is questionable if the conclusion in this paper is generalizable to other LLMs. The authors should provide experiments with other LLMs such as Flant5-xxl and Llama to justify this. On the other hand, this proposed method is based on frozen visual encoders and only the adapter layers are tuned. How about the performance of jointly tuning the visual encoders? Can the low-level visual feature naturally merge in higher layers of the visual encoder by simply fine-tuning it?


-	Some points claimed in this paper requires further justification or explanation:

o	This work claims the CLIP visual encoder lacks pixel-level information such as colour and positions. Can the proposed method solve this issue? Experimental results on segmentation-based tasks may help to justify this.

o	What are the value of the learned alphas and betas? Are feature in all layers useful for the model?

o	Are all patch features from visual encoder fed into LLM? If yes, this may result in a significantly higher number of tokens than the text side, which leads to higher memory cost and slower inference speed than other methods [1], especially the images are of 336x336 resolution.



> [1] Dai et al. InstructBLIP: Towards General-purpose Vision-Language Models with Instruction Tuning. NeurIPS 2023

> [2] Bai et al. Qwen-VL: A Versatile Vision-Language Model for Understanding, Localization, Text Reading, and Beyond. 2308.12966

> [3] Bai et al. TouchStone: Evaluating Vision-Language Models by Language Models. Arxiv 2308.16890

> [4] Li et al. SEED-Bench: Benchmarking Multimodal LLMs with Generative Comprehension. Arxiv 2307.16125

> [5] Liu et al. Visual Instruction Tuning. NeurIPS 2023

> [6] Want et al. What Makes for Good Visual Tokenizers for Large Language Models? Arxiv 2305.12223

> [7] Xiao et al. CLIP-VG: Self-paced Curriculum Adapting of CLIP for Visual Grounding. TMM 2023

**Questions:**

Please see the Weaknesses.

---

### Official Review · Reviewer_dgTi · 2023-10-30

**Soundness:** 2 fair
**Presentation:** 2 fair
**Contribution:** 2 fair
**Rating:** 3
**Confidence:** 3

**Summary:**

This paper presents an extensive investigation into different visual encoders for MLLMs. Four typical visual foundation models are considered, including CLIP, DINOv2, MAE, and DeiT. The performance is evaluated on vision-language tasks including visual grounding, object hallucination, visual question answering, image captioning, and MME benchmark. Results show that DINOv2 and CLIP should be combined with the proposed structure. Reasonable results are reported.

**Strengths:**

1. Interesting discoveries: MAE and DeiT perform inferiorly as visual branches for MLLMs
2. This paper proposes a useful approach to combine DINO and CLIP and the effectiveness is validated.

**Weaknesses:**

1. Some claims are exaggerated or lack theoretical or empirical support. For example,
"Since the language models have succeeded in scaling up the model size with progressively powerful language abilities, the short plate of the Buckets Effect for MLLMs lies in the visual models, which fails to perform in-context learning, and suffer from domain gap and limited zero-shot ability."

Does every single MLLM fail to perform in-context learning? Do they all fail because of domain gap and limited zero-shot ability?

2. This submission looks more like a technical blog than an academic paper. The discoveries are nothing amazing, which include
1) DINOv2 has fine-grained visual information. This is totally expected.
2) CLIP lacks fine-grained recognition capacity. Everyone knows this.
3) MAE and DeiT should not be used as visual encoders. Of course, because they are neither trained with text-image pairs nor DINO-style supervision.

3. The proposed structure is straightforward but, to be honest, trivial. The effect is inconsistent across tasks. For example, CLIP+MFM reduces the performance on Flickr30k (80.7 -> 79.3), makes no difference on Avg POPE, but improves VQA.

4. The code repository is empty.

**Questions:**

This paper seems to claim its contribution as 1) a systematic empirical study and 2) a useful universal fundamental visual encoder.

Though I do not think the former is interesting, the latter may be acceptable if it is shown that the proposed encoder structure + some other MLLMs can also bring consistent improvements without carefully customizing the encoder. The reported results merely show that the proposed structure suits the Shikira well, which is unconvincing and strongly indicates that the resultant model has been highly engineered.

---

### Official Review · Reviewer_EbaU · 2023-11-03

**Soundness:** 3 good
**Presentation:** 3 good
**Contribution:** 3 good
**Rating:** 6
**Confidence:** 4

**Summary:**

In this paper, authors try to use multi-level vision features and two vision backbones (CLIP and DINO) too boost the performance of current large multimodal models. Authors evaluate the model upon a set of benchmarks such as REC, REG, POPE, VQA, etc.

**Strengths:**

1. The proposed study across the features merging and assemble strategy is extensive.
2. The paper is easy to read, writing is good. Also figures.
3. Evaluations are generally extensive enough.
4. The findings are interesting, which could potentially motivate futural study on visual backbone.

**Weaknesses:**

1. The training time is pretty intensive, 120 hour on 8 NVIDIA A800 GPUs.
2. Authors mainly study how to ensemble the pretrained features, yet data and training/finetuning pipeline keeps the same.

**Questions:**

1. In Figure 1, it is hard to say whether one has more semantic or low-level visual features. Since the deep layers in CLIP and shallow layers in DINO looks similar in visualization, I can also say they both share high level semantics.
2. Equ.1 error appears.
3. Does author needs a pretraining stage for multimodal connector, such as those in LLaVA and Mini-GPT4? In this paper seems like there is no such part. Is directly training LLMs more efficient/effective than conducting a pretraining stage first?

---

### Meta-Review · Area_Chair_cNQc · 2023-12-11

**Metareview:**

This paper probes the visual backbones of MLLMs. Specifically, it explores the integration of CLIP and DINO for enhancing MLLMs, with performance assessment across several benchmarks. While the reviewers find this paper interesting to read, they unanimously raise several major concerns about the methodology, novelty, and overall contribution. Unfortunately, the authors have not provided a rebuttal addressing these concerns.

The AC encourages the authors to carefully tackle these concerns and make a stronger submission next time.

**Justification For Why Not Higher Score:**

The authors of this paper did not provide a rebuttal to address the concerns raised by reviewers.

**Justification For Why Not Lower Score:**

N/A

---

### Decision · Program_Chairs · 2024-01-16

Reject